evolution/mathematical modelling/complexity

cultural evolution, paradigm shift, epistasis, cognitive dissonance, irreversibility, phase transition

**Author for correspondence:**
José A. Cuesta
e-mail: cuesta@math.uc3m.es

# Epistasis between cultural traits causes paradigm shifts in cultural evolution

Ignacio Pascual[1,2], Jacobo Aguirre[1,3], Susanna Manrubia[1,3] and José A. Cuesta[1,2,4,5]

[1]Grupo Interdisciplinar de Sistemas Complejos (GISC), Madrid, Spain
[2]Departamento de Matemáticas, Universidad Carlos III de Madrid, Spain
[3]Centro Nacional de Biotecnología (CNB-CSIC), Madrid, Spain
[4]Instituto de Biocomputación y Física de Sistemas Complejos (BIFI), Universidad de Zaragoza, Zaragoza, Spain
[5]UC3M-BS Institute of Financial Big Data (IFiBiD), Madrid, Spain

SM, 0000-0003-0134-2785; JAC, 0000-0001-9890-9367

Every now and then the cultural paradigm of a society changes. While current models of cultural shifts usually require a major exogenous or endogenous change, we propose that the mechanism underlying many paradigm shifts may just be an emergent feature of the inherent congruence among different cultural traits. We implement this idea through a population dynamics model in which individuals are defined by a vector of cultural traits that changes mainly through cultural contagion, biased by a 'cultural fitness' landscape, between contemporary individuals. Cultural traits reinforce or hinder each other (through a form of cultural epistasis) to prevent cognitive dissonance. Our main result is that abrupt paradigm shifts occur, in response to weak changes in the landscape, only in the presence of epistasis between cultural traits, and regardless of whether horizontal transmission is biased by homophily. A relevant consequence of this dynamics is the irreversible nature of paradigm shifts: the old paradigm cannot be restored even if the external changes are undone. Our model puts the phenomenon of paradigm shifts in cultural evolution in the same category as catastrophic shifts in ecology or phase transitions in physics, where minute causes lead to major collective changes.

## 1. Background

We live in a quantitative world. We are so deeply used to measuring everything in and around us that it is difficult to imagine it may have been otherwise. However, quantitative societies are relatively recent happenings. In his book *The*

*Measure of Reality* [1], historian Alfred W. Crosby explains that in the Middle Ages Europeans did not pay much attention to time. Their qualitative way of thinking provided a coherent and sufficient model of the world, even if dates were not very precise or the day was divided in twelve hours from dawn till sunset, regardless of whether it was winter or summer. By 1250, new external pressures (such as the rise of the European population, the migration of peasants to cities, the flourishing of commerce with new, distant markets) started to question the qualitative model. But, actually, it was the acquisition of quantitative habits in marginal aspects of culture (accurate time measure in music, geometric description in painting, bookkeeping in business management, etc.) that eventually drove the change. In the *cultural paradigm shift* that took place in the transition from the Middle Ages to the Renaissance, culture drastically changed in the turn of a few generations. Kuhn, who coined the term 'paradigm shift', proposed a similar mechanism to explain scientific revolutions [2].

Some remarks are worth pointing out. First, Crosby's essay suggests that paradigm shifts are not limited to the dynamics of science, but can be found in more general cultural settings (arts, fashion, cooking, laws, philosophy, etc.). Second, they can be thought of as an evolutionary phenomenon— there is a change in the cultural paradigm in response to a change of the 'environment' (understood in a broad sense). Third, the presence of some cultural elements affects the relative importance of other cultural elements in the individuals' cultural state. And fourth, the paradigm shift is an abrupt phenomenon in historical time scale (i.e. compared to the lifetime of each paradigm)—for instance, the prehistoric archaeological record reveals long periods where tools hardly change, which are 'suddenly' replaced by completely different toolkits, full of new, more efficient, even more diverse tools [3,4]. The first two points bring the topic of cultural paradigm shifts into the domain of cultural evolution [5]; the third one aligns with recent work emphasizing the importance of cultural elements as enhancers or inhibitors of other cultural elements [6]; the last point resembles the concept of punctuated equilibrium in biology [7,8] (not unbeknown to cultural evolution [9–13]), or of critical phenomena in physics—where small changes in external parameters induce abrupt changes of measurable magnitudes [14].

There has been a long debate in the literature about the origins of cultural paradigm shifts, and different models, resorting to different causes, have been proposed. Some of these models stress the importance of externally forced, fast environmental [10] or cognitive [9] changes. In others, those changes are endogenous—like the appearance of innovative breakthroughs [11–13]. The bottom line of all these models though is that major changes demand major causes. It is out of the question that any of these causes must have played a role in many paradigm shifts along human history. However, there are plenty of unforeseeable cultural shifts (e.g. the transition from the Middle Ages to the Renaissance; the change of the narrative paradigm at the beginning of the twentieth century; the sudden emergence of new fashions in spoken language; the change of the moral attitudes or religious beliefs at the turn of one or a few generations; etc.), which cannot be explained by any of these reasons. They do not seem to require an environmental or cognitive change; they do not appear to be the consequence of a serendipitous breakthrough. They seem to fit better in Kuhn's or Crosby's scheme—an accumulation of small changes eventually bringing about a cultural revolution.

Although biological and cultural evolution do not share the same microscopic mechanisms [5], they are deeply related [15], and often, the former has inspired the latter. Many models of cultural evolution are suitable adaptations of those of population genetics, incorporating variants of the standard mechanisms of replication, mutation and drift [16,17], but also—building on [6]—of branching and recombination [18]. Likewise, the concept of epistasis in genetics (i.e. the mutual dependence between two genes or two positions in a sequence) also has its counterpart: cultural epistasis has been used to refer to the association between two ideas due to the existence of a logical consequence in their contents [19].

As it turns out, certain models of evolution of heterogeneous populations in varying environments show that epistasis between the *loci* of a molecular sequence cause abrupt changes in the composition of the population under smooth environmental changes [20,21]. Hence cultural epistasis seems like a promising root to look for a microscopic explanation of paradigm shifts. As a matter of fact, it has been suggested that cultural traits may act as facilitators or inhibitors of other traits in modelling the appearance and accumulation of innovations [6,11]. The idea that traits interact with each other holds in a wider context though. Language evolution is driven by interaction of its specific traits. For instance, although $n$ and $m$ are phonetically distinct, the presence of a subsequent $p$ inhibits the $n$ in favour of the $m$ [22]. Semantics is strongly affected by a network of close concepts, to the point that the meaning of a word can shift as a consequence of a change in this network [23]. Also, the acquisition of additional languages is facilitated by prior knowledge of two or more languages, and

brings about effects in other aspects of the individuals' personal lives [24]. Other examples of interaction between cultural traits are the correlation between right-wing authoritarianism belief and low openness to experience [25], religious beliefs and health practices [26], or animal ethical profiles and diet choices [27]. In the examples above, the consistency of the overall cultural state entails that either both or none of the traits are present with higher likelihood. This correlation between traits can be cast as a form of epistasis known as reciprocal sign epistasis, which is a necessary condition for a multi-peaked fitness landscape [28]. Also, incompatibilities between traits that lead to zero-fitness states yield qualitatively equivalent landscapes.

In this work, we introduce a population dynamics model that implements common mechanisms of cultural transmission [16,17,29,30]. The model has the novelty of incorporating a microscopic mechanism of trait interaction (cultural epistasis) and imperceptible variations in the social value of cultural states. The results reveal that, indeed, cultural epistasis may underlie many paradigm shifts that are difficult to explain otherwise, since no major exogenous or endogenous changes were involved in their emergence. In the light of these results, we interpret these paradigm shifts as emergent features of the tension between the different cultural traits shared by a population—thus bringing the phenomenon to the realm of phase transitions and critical phenomena.

## 2. Model

### 2.1. Definition and notation

The culture of an individual, defined as the information acquired from other individuals via social transmission [5] is defined by a set of beliefs, attitudes, preferences, knowledge, skills, customs and norms. In an abstract model of culture, every individual can be represented by an array of cultural attributes, each having one out of a set of possible values [31]. To keep things simple we will assume that each of these attributes can be determined by a yes/no question (e.g. 'are you a Christian?', 'do you like jogging?', 'do you eat chocolate?', 'do you speak English?') and so it can take only two values—say 0 or 1. Thus, *cultural states* are vectors $\mathbf{s} = (s_1, \ldots, s_n)$, with $s_i \in \{0, 1\}$. Distance between cultural states $\mathbf{s}$ and $\mathbf{s}'$ will be measured as the number of different attributes (Hamming distance) and denoted $d_H(\mathbf{s}, \mathbf{s}')$. At a given time $t \geqslant 0$, the fraction of the population in cultural state $\mathbf{s}$ will be denoted $x(\mathbf{s}, t)$. Population will be assumed very large and constant—so that demographic fluctuations are negligible.

For later convenience, we will sometimes denote $\mathbf{s} = (s_i, \mathbf{s}_{-i})$, separating the $s_i$ component out of vector $\mathbf{s}$ and gathering the remaining $n-1$ components in $\mathbf{s}_{-i}$. Also, we will use the short-hand $\bar{s}_i = 1 - s_i$.

Rather than assuming an intrinsic adaptive value to the different attributes of a cultural vector—as it is often assumed in models of cultural evolution [16,17]—we will assign a *fitness* $F(\mathbf{s})$ to the whole cultural state $\mathbf{s}$. Here, fitness is understood as a measure of the internal consistency of the set of cultural attributes forming that state, which eventually determines how 'happy' an individual is in cultural state $\mathbf{s}$ and how prone she is to adopt alternative traits—the higher the fitness, the more reluctance to change. From a psychological perspective, a low fitness can be associated with the cognitive dissonance caused by the coexistence of conflicting traits in the cultural state of one individual [32].

### 2.2. Fitness landscapes, epistasis and environmental changes

In a realistic fitness landscape, the nature and strength of interactions between cultural traits will depend on which specific traits are involved. In the absence of specific data in this respect, the simplest approach is analogous to that used in models of biological evolution to obtain (rough) epistatic landscapes. Specifically, we will use Kauffman's NK landscape [33] (see below for details). This model has two parameters: the number of traits $n$, and the degree of epistasis $k$. If $k = 0$, traits contribute additively and independently to fitness. If $k > 0$, changing a trait affects the contribution of other $k$ traits to the fitness.

Incompatibilities among traits—e.g. risk-averse people do not practise paragliding—lead to zero-fitness states. Since by construction the NK model yields $F(\mathbf{s}) > 0$ for any $\mathbf{s}$, following [20] we have introduced a small change in the construction of $F(\mathbf{s})$ to account for incompatibilities. We define a threshold value $f_{th}$ and redefine

$$F_{th}(\mathbf{s}) = \begin{cases} F(\mathbf{s}) - f_{th}, & \text{if } F(\mathbf{s}) > f_{th}, \\ 0, & \text{if } F(\mathbf{s}) \leqslant f_{th}. \end{cases} \qquad (2.1)$$

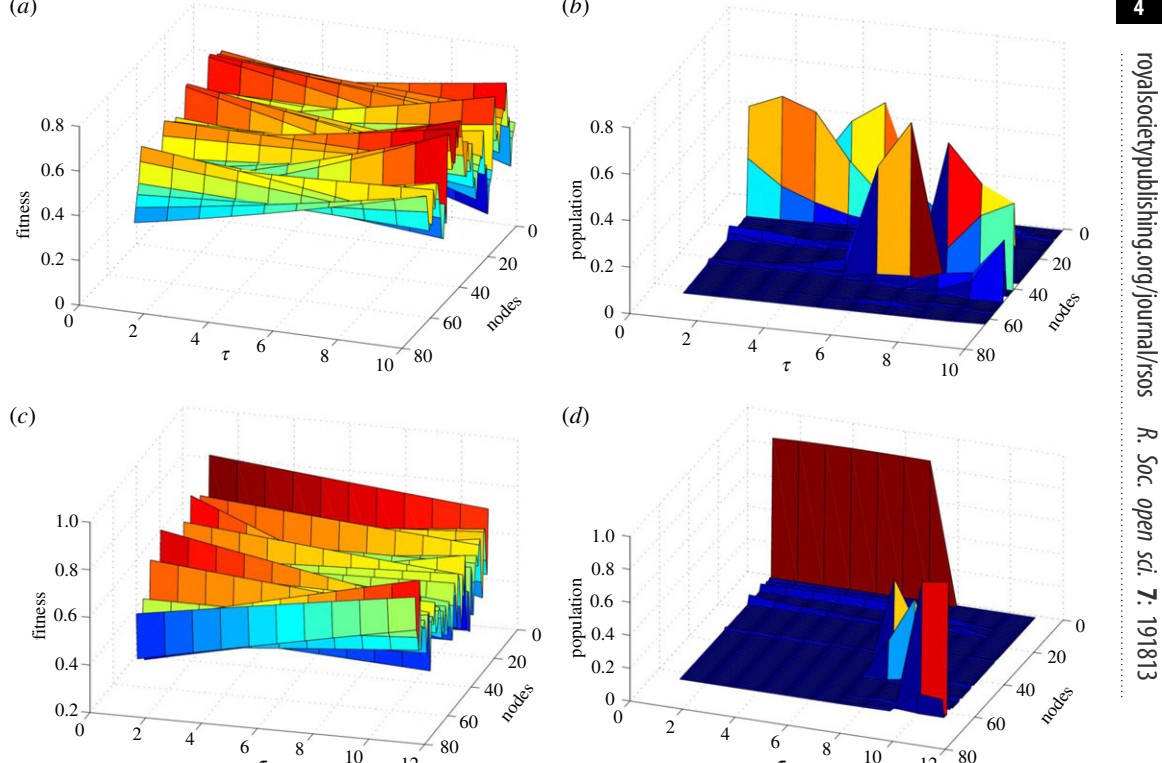

**Figure 1.** Example fitness landscapes and evolution of the population density when $\tau$ grows. (*a,c*) Fitness values of the landscape for weak and strong epistasis, $k = 1$ and $k = 5$, respectively. (*b,d*) The density of population at each node once equilibrium has been reached for the landscapes in (*a*) and (*c*), respectively, and for each value of $\tau$. While the fitness landscapes undergo a linear and smooth variation, the populations show a nonlinear evolution, especially drastic in the latter case. In both examples, $f_{th} = 0$, $\beta = 1$, $N = 6$, $\varepsilon = 0.1$.

Accordingly, all cultural states for which $F(\mathbf{s}) \leqslant f_{th}$ have zero fitness: no individual bears such a combination of traits.

Paradigm shifts occur as a response to parsimonious 'environmental' change. External influences may change the epistatic interaction between the different traits, leading to a modification of the fitness landscape, and where the region of overall higher fitness is displaced [34], see also the representation in figure 1. Imperceptible—but also unavoidable—exogenous changes are implemented by constructing two different landscapes, $F_0(\mathbf{s})$ and $F_1(\mathbf{s})$ (for example, generating two independent samples of an NK landscape with the same parameters, see below), and then defining the fitness landscape as the convex combination $F_\tau(\mathbf{s}) = \tau F_1(\mathbf{s}) + (1 - \tau)F_0(\mathbf{s})$. As $\tau$ varies from 0 to 1, $F_\tau(\mathbf{s})$ varies continuously from $F_0(\mathbf{s})$ to $F_1(\mathbf{s})$. Figure 1*a,c* shows two examples of variations in the fitness of each node in the landscape for $k = 1$ and $k = 5$, respectively. Note that fitness varies parsimoniously at each node, as implied by the linear interpolation above.

## 2.3. Kauffman's NK landscape

In Kauffman's NK landscape, the fitness corresponding to the vector $\mathbf{s}$ is calculated as

$$F(\mathbf{s}) = \frac{1}{n} \sum_{i=1}^{n} \phi_i(s_i, s_{[i+1]}, \ldots, s_{[i+k]}), \tag{2.2}$$

where $[j] = j$ if $j \leqslant n$ and $[j] = j - n$ if $j > n$. Here $\phi_i(\sigma_1, \ldots, \sigma_{k+1})$, for $i = 1, \ldots, n$, stand for $n$ different functions of $k + 1$ Boolean variables each. The $2^{k+1}$ values that each of these functions can take are picked randomly from a uniform distribution. When $k = 0$ each attribute makes an additive contribution to the fitness (no epistasis). When $k = n$ the model defines a random landscape (maximal epistasis). That definition of fitness includes reciprocal sign epistasis, which generates multi-peaked landscapes with more than one maximum.

With the exception of the example depicted in figure 1, the parameter $\tau$ defining the intermediate fitness landscapes $F_\tau(\mathbf{s})$ varies between 0 and 1 in steps of $\varepsilon = 0.01$.

## 2.4. Dynamics

Cultural transmission in this model will be assumed horizontal (peer-to-peer). The mechanisms through which horizontal transmission occurs have been much debated. A common assumption is homophily, that is, the more similar our peers, the more they influence us [31,35]. However, it seems that some attributes (e.g. religion, political beliefs, social status [35]) are more prominent than others when we seek for similarities with someone. For instance, links in the blogosphere are made almost exclusively between blogs of the same political sign [36], even though their authors may differ in many other cultural traits. On the other hand, some of the strongest cultural influences we may receive come from books, whose authors may be entirely unknown to us except for those features revealed by the arguments they deploy. Often we change our mind about some issue after a discussion with other people—which sometimes we only witness, as in the case of TV debates—on that specific topic. What is important about these interactions is that we are more prone to change one cultural trait if the cultural state we end up with is globally more coherent—more capable to cope with reality—and has therefore a higher fitness.

For all these reasons, we will assume a simple dynamics in which individuals meet in pairs and put a random cultural attribute at stake. These meetings may be biased by homophily. If both individuals disagree in that attribute either of them can change her trait according to the difference between her current fitness and the fitness of her cultural state after the change. The probability that someone with cultural state $\mathbf{s}'$ adopts cultural state $\mathbf{s}$ will be modelled as

$$\Pr\{\mathbf{s}' \to \mathbf{s}\} = \mathcal{G}\left(\frac{F(\mathbf{s})}{F(\mathbf{s}')}\right), \tag{2.3}$$

where $\mathcal{G}(x)$ is a sigmoid function such that $\mathcal{G}(0) = 0$, $\mathcal{G}(x) \to 1$ as $x \to \infty$, and $\mathcal{G}(1) = 1/2$. The choice

$$\mathcal{G}(x) \equiv \frac{x^\beta}{1 + x^\beta} \tag{2.4}$$

allows us to tune how sharp it goes from 0 to 1 as $x$ crosses 1 by selecting an appropriate $\beta > 0$. A large value of $\beta$ makes $\mathcal{G}(x) \approx 1$ for almost all $x > 1$ (i.e. $F(\mathbf{s}) > F(\mathbf{s}')$), and a smaller value of $\beta$ makes $\mathcal{G}(x)$ smoother, showing some reluctance to change even though $x > 1$, but also giving some probability of changing even if $x < 1$. Thus, $\beta$ measures individuals' discomfort towards cognitive dissonances—the larger $\beta$ the more prone they are to adopt traits that increase internal consistency relieving cognitive dissonances.

The fact that $\mathcal{G}$ is a function of the fitness ratio allows us to normalize all fitness values without losing generality. So fitness will be forced to be $0 \leqslant F(\mathbf{s}) \leqslant 1$ for all $\mathbf{s} \in \{0, 1\}^n$.

Changes through meetings will, therefore, occur at a rate

$$R_m(\mathbf{s}' \to \mathbf{s}, t) = \lambda x(\mathbf{s}', t)x(\mathbf{s}, t)\mathcal{W}(\mathbf{s}', \mathbf{s})\mathcal{G}\left(\frac{F(\mathbf{s})}{F(\mathbf{s}')}\right), \tag{2.5}$$

where $\lambda x(\mathbf{s}', t)x(\mathbf{s}, t)$ is the rate of pairwise meetings and

$$\mathcal{W}(\mathbf{s}', \mathbf{s}) \equiv \frac{d_H(\mathbf{s}', \mathbf{s})}{n}\left(1 - \frac{d_H(\mathbf{s}', \mathbf{s})}{n}\right)^\alpha \frac{(1 + \alpha)^{1+\alpha}}{\alpha^\alpha}. \tag{2.6}$$

In this function, the factor $d_H(\mathbf{s}', \mathbf{s})/n$ is the probability that the two individuals differ in a randomly chosen attribute, whereas $[1 - d_H(\mathbf{s}', \mathbf{s})/n]^\alpha$ weights the influence of homophily—the more so the larger $\alpha$. The last numerical factor is there to ensure that the largest value of $\mathcal{W}(\mathbf{s}', \mathbf{s})$, as a function of $d_H(\mathbf{s}, \mathbf{s}')$, is 1. This maximum is reached for $d_H(\mathbf{s}, \mathbf{s}') = n/(1 + \alpha)$. Larger values of $\alpha$ therefore put the most influential people—here understood as the individuals most able to produce a change of trait upon an encounter—at smaller Hamming distances. In other words, increasing $\alpha$ favours homophily.

On top of that we also introduce the possibility of spontaneous changes of mind in single traits. Their rate will be

$$R_i(\mathbf{s}' \to \mathbf{s}, t) = \mu x(\mathbf{s}', t)\mathcal{G}\left(\frac{F(\mathbf{s})}{F(\mathbf{s}')}\right). \tag{2.7}$$

The last factor in this expression introduces bias in the adoption of new traits: in general, only those traits that increase fitness will have a chance to spread. Overall, $R_i(\mathbf{s}' \to \mathbf{s}, t)$ implements the concept of guided variation, whereby individuals try to improve their fitness through trial and error by tinkering with the traits they have inherited [17].

With these elements, the dynamic equation that balances the flux of individuals in and out of a cultural state $\mathbf{s}$ is

$$\frac{d}{dt} x(\mathbf{s}, t) = \mathcal{I}(\mathbf{s}, t) - \mathcal{O}(\mathbf{s}, t), \tag{2.8}$$

where

$$\mathcal{I}(\mathbf{s}, t) = \sum_{i=1}^{n} \left[ \lambda \sum_{\mathbf{z}_{-i}} \mathcal{W}[(\bar{s}_i, \mathbf{s}_{-i}), (s_i, \mathbf{z}_{-i})] x(s_i, \mathbf{z}_{-i}, t) + \mu \right] x(\bar{s}_i, \mathbf{s}_{-i}, t) \mathcal{G}\left( \frac{F(\mathbf{s})}{F(\bar{s}_i, \mathbf{s}_{-i})} \right) \tag{2.9}$$

and

$$\mathcal{O}(\mathbf{s}, t) = \sum_{i=1}^{n} \left[ \lambda \sum_{\mathbf{z}_{-i}} \mathcal{W}[\mathbf{s}, (\bar{s}_i, \mathbf{z}_{-i})] x(\bar{s}_i, \mathbf{z}_{-i}, t) + \mu \right] x(\mathbf{s}, t) \mathcal{G}\left( \frac{F(\bar{s}_i, \mathbf{s}_{-i})}{F(\mathbf{s})} \right). \tag{2.10}$$

Internal sums run over all choices of $\mathbf{z}_{-i} \in \{0, 1\}^{n-1}$. The interaction dynamics that this equation reflects assumes two things: first, that every two individuals have the same chance to meet (well-mixed population), and second, that at every encounter only one trait is susceptible to change—the other ones being irrelevant. As a result, the cultural state of an individual can only experience gradual changes, one trait at a time. Since the number of different traits remains constant along the process, our model does not include innovation in the sense of enlarging the cultural repertoire, as most models assume [11–13].

To illustrate the response of the population evolving under the dynamics just described to parsimonious environmental changes, we represent in figure 1b,d the variation in the cultural states of a population subjected to the landscapes depicted in figure 1a,c. In that figure, we have varied $\tau$ in steps of $\varepsilon = 0.1$ for illustrative purposes. From here onward, the parameter $\tau$ is assumed to change very slowly with time—so much so that the system has enough time to reach the steady state before $\tau$ changes appreciably. On the other hand, trait incompatibilities are assumed to be independent of environmental changes, so in the model with $f_{\mathrm{th}} > 0$, all zero-fitness states are maintained for all $0 \leqslant \tau \leqslant 1$.

## 2.5. Numerical integration

We integrate the differential equations (2.8)–(2.10) using a fourth-order Runge–Kutta method [37]. Since population densities lay in the interval $0 \leqslant x(\mathbf{s}) \leqslant 1$, to avoid numerical errors due to small values of $x(\mathbf{s})$ we have rewritten the equations in terms of the variables $y(\mathbf{s}) = \log x(\mathbf{s})$ as $\dot{y}(\mathbf{s}) = \mathcal{I}'(\mathbf{s}, t) - \mathcal{O}'(\mathbf{s}, t)$, where

$$\mathcal{I}'(\mathbf{s}, t) = \sum_{i=1}^{n} \left[ \lambda \sum_{\mathbf{z}_{-i}} \mathcal{W}[(\bar{s}_i, \mathbf{s}_{-i}), (s_i, \mathbf{z}_{-i})] e^{y(s_i, \mathbf{z}_{-i}, t)} + \mu \right] e^{y(\bar{s}_i, \mathbf{s}_{-i}, t) - y(\mathbf{s}, t)} \mathcal{G}\left( \frac{F(\mathbf{s})}{F(\bar{s}_i, \mathbf{s}_{-i})} \right)$$

and

$$\mathcal{O}'(\mathbf{s}, t) = \sum_{i=1}^{n} \left[ \lambda \sum_{\mathbf{z}_{-i}} \mathcal{W}[\mathbf{s}, (\bar{s}_i, \mathbf{z}_{-i})] e^{y(\bar{s}_i, \mathbf{z}_{-i}, t)} + \mu \right] \mathcal{G}\left( \frac{F(\bar{s}_i, \mathbf{s}_{-i})}{F(\mathbf{s})} \right).$$

We have used an integration time-step $\Delta t = 0.1$, and run the integration method until the maximum difference between $x(\mathbf{s}, t)$ and $x(\mathbf{s}, t + 100)$ is smaller than $10^{-4}$.

The procedure we implement for each value of $\tau$ that interpolates between the initial and the final landscape goes as follows. We start off from the fitness landscape $\mathbf{F}_0$ and solve the equation starting from a uniform initial condition until an equilibrium is reached. We then increase $\tau$ by a small amount and solve again the equations, taking the equilibrium population vector previously obtained as the initial condition for this new fitness landscape $\mathbf{F}_\tau$, until we reach a new equilibrium vector. We iterate until the final landscape $\mathbf{F}_1$ is reached.

## 2.6. Model parameters

Cultural vectors with $n = 6$ traits will be used in our simulations. They correspond to 64 different cultural 'states', numbered from 0 to 63 according to the decimal expression of their binary representation (e.g. 5 = 000101, 37 = 100101). We assume that in 99.9% of the cases changes come about

through transmission, and in the remaining 0.1% they occur through spontaneous changes (i.e. $\lambda = 0.999$, $\mu = 0.001$). Different levels of epistasis are tested by varying $k$ and $f_{th}$. The parameter $\beta$ is varied as well. Unless otherwise stated we set $\alpha = 0$, meaning absence of homophily.

## 2.7. Similarity measure

For vectors $\mathbf{x}$ with components $x(\mathbf{s})$, $\mathbf{s} \in \{0, 1\}^n$, we introduce a measure of similarity that takes into account not only how different two vectors are but also the Hamming distance between their most prominent components. This is achieved through the inner product

$$\langle \mathbf{x}, \mathbf{y} \rangle = \sum_{\mathbf{s}, \mathbf{s}' \in \{0,1\}^n} x(\mathbf{s}) \big[ n - d_H(\mathbf{s}, \mathbf{s}') \big] y(\mathbf{s}'). \tag{2.11}$$

Similarity is then computed as

$$\text{sim}(\mathbf{x}, \mathbf{y}) = \frac{\langle \mathbf{x}, \mathbf{y} \rangle}{\sqrt{\langle \mathbf{x}, \mathbf{x} \rangle \langle \mathbf{y}, \mathbf{y} \rangle}}. \tag{2.12}$$

By construction, $\text{sim}(\mathbf{x}, \mathbf{y}) \leqslant 1$, and the largest similarity is achieved when $\mathbf{x} = \mathbf{y}$. Also if $x(\mathbf{s})$, $y(\mathbf{s}) \geqslant 0$ for all $\mathbf{s} \in \{0, 1\}^n$ then $\text{sim}(\mathbf{x}, \mathbf{y}) \geqslant 0$, and 0 is achieved when only one component of both vectors is non-zero, say $x(\mathbf{s})$ and $y(\mathbf{s}')$, and $d_H(\mathbf{s}, \mathbf{s}') = n$. Thus $\text{sim}(\mathbf{x}, \mathbf{y})$ quantifies not just how different vectors $\mathbf{x}$ and $\mathbf{y}$ are—as the ordinary dot product does—but also if differences occur in components that are close or far from each other. To understand the importance of the latter, consider the case where $n = 2$, so the only four cultural states are (0, 0), (0, 1), (1, 0) and (1, 1), which we order as 0, 1, 2 and 3, respectively. With the standard dot product, if $\mathbf{x} = (1, 0, 0, 0)$, $\text{sim}(\mathbf{x}, \mathbf{y}_i) = 0$ for $\mathbf{y}_1 = (0, 1, 0, 0)$, $\mathbf{y}_2 = (0, 0, 1, 0)$ or $\mathbf{y}_3 = (0, 0, 0, 1)$. However, with our similarity measure, $\text{sim}(\mathbf{x}, \mathbf{y}_1) = \text{sim}(\mathbf{x}, \mathbf{y}_2) = 1/2$ and $\text{sim}(\mathbf{x}, \mathbf{y}_3) = 0$. In other words, $\mathbf{y}_1$ or $\mathbf{y}_2$ are considered less dissimilar to $\mathbf{x}$ than $\mathbf{y}_3$ because the 1 component is closer to that of $\mathbf{x}$ than is the 1 component of $\mathbf{y}_3$.

# 3. Results

## 3.1. Paradigm shifts are a consequence of epistasis

Equation (2.8) has been numerically solved (see Model section) in two situations where the landscape ignored ($k = 0$) or included ($k > 0$) epistatic interactions between cultural traits. Along this process we monitor the similarity between the initial and current population vectors $\text{sim}(\mathbf{x}_0, \mathbf{x}_\tau)$. For this and all other cases that we will discuss, $\text{sim}(\mathbf{F}_\tau, \mathbf{F}_{\tau - \Delta\tau}) \approx 1$ all along $0 \leqslant \tau \leqslant 1$, meaning that changes in the landscape are barely noticeable.

Still, changes in populations are important and qualitatively different depending on epistasis, as figure 2 illustrates. If epistasis is absent (figure 2a), the population vector undergoes a big change as $\tau$ ranges from 0 to 1, meaning that the bulk of the population has changed location in the cultural landscape, reaching a final state different from the initial one. However, the curve depicted is smooth, implying that the cultural change is continuous. By contrast, even the mildest amount of epistasis may induce abrupt changes in the population in response to weak changes in the environment. For landscapes with $k = 1$ and $f_{th} = 0$, the similarity of the population vectors can undergo discontinuous changes such as those plotted in figure 2b. The introduction of incompatibilities in the form $f_{th} > 0$ can be interpreted as an extreme form of epistasis (analogous to synthetic lethality in genomics [38]), where two traits, non-lethal by themselves, cannot be combined into a viable state. As figure 2c shows, also this form of epistasis leads to discontinuities in the cultural states under mild environmental changes. These two forms of epistasis differ in their origin, since increases in $k$ map into increased roughness of the landscape, while a threshold $f_{th} > 0$ reflects the level of tolerance to cognitive dissonance. Still, they affect in similar ways the topology of the fitness landscape by increasing the heterogeneity of contacts between cultural states and causing bottlenecks to movements in the fitness landscape, either in the form of fitness valleys or of narrow pathways, in both cases to avoid fitness losses.

In figure 3 we show the configuration space $\{0, 1\}^n$ as a network, where each node represents one cultural vector (out of 64) and whose links connect nodes at Hamming distance 1 (nearest neighbours, differing in one cultural trait). As the fitness landscape changes, the node with the largest value of fitness may gradually change. Two examples, corresponding to landscapes in figure 2a,b are shown.

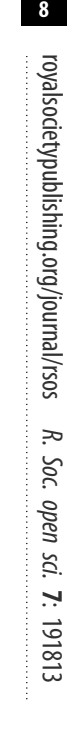

**Figure 2.** Similarity between the population vectors $\mathbf{x}_\tau$ and $\mathbf{x}_0$ as a function of $\tau$; $\beta = 1$. (*a*) The landscape has no epistasis whatsoever ($k = 0$, $f_{th} = 0$). The population vector undergoes a large but smooth variation as the landscape changes. (*b*) The landscape is weakly epistatic ($k = 1$, $f_{th} = 0$). The jump discontinuity observed around $\tau = 0.79$ reveals an abrupt change in the cultural composition of the population. (*c*) Epistasis is introduced in the landscape through trait incompatibility ($k = 0$, $f_{th} > 0$, chosen so that the landscape has 24 incompatible states). The dynamics are qualitatively equivalent to landscapes with $k > 0$.

They illustrate smooth and discontinuous changes in the cultural state of the population, both under mild environmental changes, when epistasis is absent or present, respectively. In this representation, the sudden transition in figure 2*c* is qualitatively analogous to that shown in figure 3*b*. In the latter case, it is of interest to note that, initially, the population does not sit at the best possible cultural state, and, in spite of that, it is very resilient to respond to environmental changes. Eventually, however, a minor change drives the population to the global maximum in a dramatic paradigm shift (five out of the six cultural traits change to their opposite values).

This section illustrates our first and main result: according to the model just introduced, paradigm shifts occur only if there is epistasis between cultural traits or, in other words, when they influence each other—so that the presence of one trait enhances or hinders the presence of another one. For the shift to emerge there is no need to add any new trait or to induce any drastic change of any kind.

## 3.2. Paradigm shifts are irreversible

The discontinuous nature of paradigm shifts has the effect that, once the shift has occurred, the old paradigm cannot be restored even if the environmental change that has produced the new paradigm

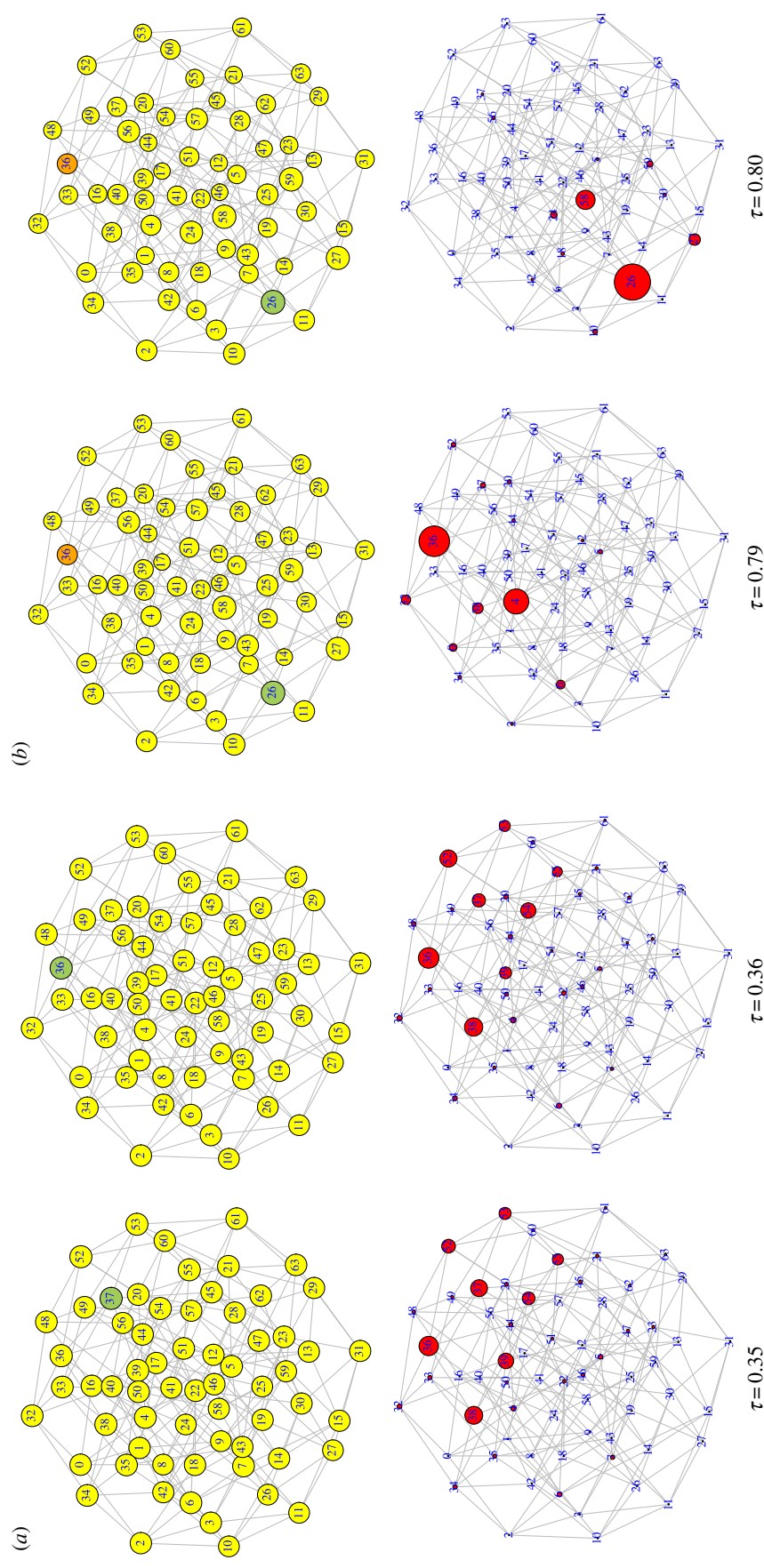

**Figure 3.** Graph representing all cultural vectors of length $n = 6$ (64 nodes). Two nodes are connected if they differ in only one trait. Above, node size is proportional to the fitness of the cultural state. A green-coloured node identifies the node with the largest fitness, while the node in orange represents a local maximum. Changes in fitness are unnoticeable in these graphs. Below, node size is proportional to the fraction of population at that cultural state. (*a*) Two equilibrium solutions for the landscapes in figure 2*a*, where $k = 0$. The node with the largest fitness changes gradually from 37 (=100101) to 36 (=100100) around the two values of $\tau$ shown. The population spreads over a few nearest neighbours of the node with the largest fitness. No abrupt changes occur during this process. (*b*) Two equilibrium solutions for the landscapes in figure 2*b*, where $k = 1$. Initially, the population sits around the local maximum at node 36 = 100100, but jumps discontinuously to the global maximum at node 26 (=011010) between $\tau = 0.79$ and $\tau = 0.80$.

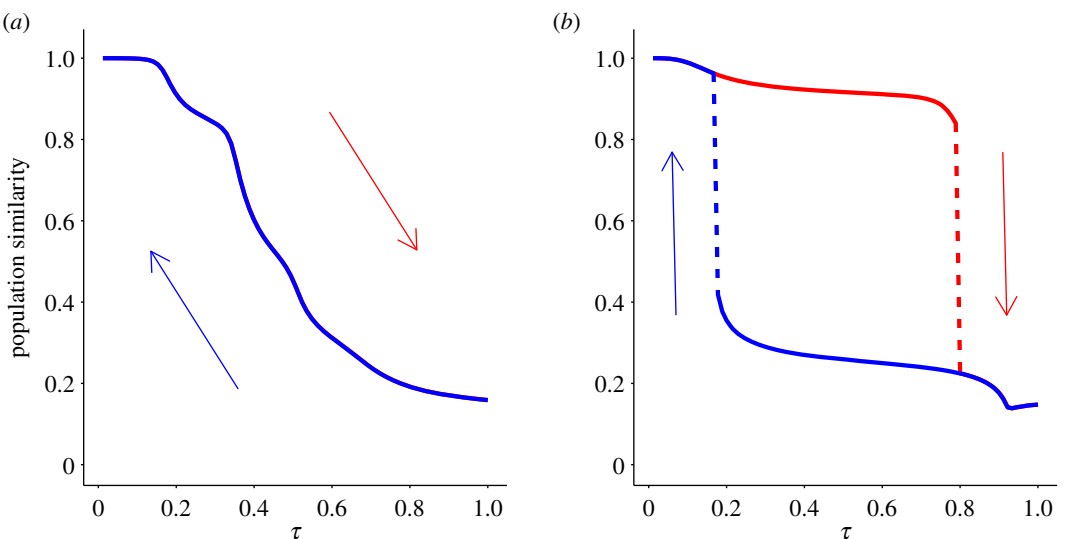

**Figure 4.** Same similarity curves as in figure 2a,b obtained both, upon increasing $\tau$ (red curve) and upon decreasing $\tau$ (blue curve). Without epistasis ($k = 0$) (a) the two curves match perfectly; with epistasis ($k = 1$), (b) a hysteresis cycle shows up as a consequence of the paradigm shift.

is reversed. In the context of dynamical systems, this behaviour is known as hysteresis, and it has been shown to occur in similar models describing sudden shifts in genomic spaces [20].

In order to illustrate this effect in cultural transitions, we have driven the system by increasing $\tau$—as described at the beginning of §3.1—and, once the paradigm shift has taken place, we drive the system back by decreasing $\tau$ down to values that it had before the transition. Figure 4 depicts the result of this process for a landscape without epistasis ($k = 0$) and for a landscape with epistasis ($k = 1$). The difference is remarkable. While the evolution in the landscape without epistasis is fully reversible (forward and backward curves are indistinguishable), the epistatic landscape induces an irreversible paradigm shift: upon decreasing $\tau$ past the tipping point the population remains in the new paradigm. We have to push $\tau$ way down this value in order to recover the old paradigm—through another abrupt paradigm shift.

## 3.3. Equilibria depend on the initial condition

The hysteresis observed around paradigm shifts indicates that equilibria in this model (stable cultural states) depend on the initial conditions, at least when the landscape is epistatic. This is not surprising given the nonlinear nature of the evolution equations, and is consistent with the intuitive idea that the current cultural state of a population somehow anchors its future evolution. However, the model differs in this respect from standard mutation-selection models which possess unique equilibria for any fitness landscape [20].

Figure 5 illustrates this effect: two close initial conditions, in which the population is distributed between two nodes (4 and 26) with slightly different proportions, end up in two very different equilibria (one with the population concentrated at 4 and the other one concentrated at 26).

## 3.4. Tolerance to cognitive dissonances hinders drastic paradigm shifts

Parameter $\beta$ tunes how resistant individuals are to adopt traits that increase the inconsistency of their cultural state. The lower $\beta$, the more inconsistencies they tolerate. All results shown so far have been obtained with $\beta = 1$. For this reference value, populations are rather focused on one or very few different cultural states. This is an indication that individuals are very reluctant to adopting inconsistent traits. If we lower this parameter, say to $\beta = 0.1$ (figure 6), the cultural heterogeneity of the population changes. Populations are more evenly spread over the network and they adapt more easily in response to environmental changes. This has at least two consequences: they are more susceptible to paradigm shifts (which happen for smaller values of $\tau$), and the corresponding discontinuous jumps are smaller. Conversely, populations which are more homogeneous (corresponding to a

**Figure 5.** Dependence of cultural states on initial conditions in epistatic landscapes. $k = 1$, $f_{th} = 0$. (a) Fitness landscape, with local maxima in orange and the absolute maximum in green. (b) and (d) depict two different initial conditions in which the population is distributed in two nodes (4 = 000100 and 26 = 011010) with almost equal concentration, but a slight bias towards one or the other. They lead to the asymptotic states (c) and (e), respectively.

higher $\beta$) are more resistant to environmental changes; however, paradigm shifts are more abrupt when they occur.

## 3.5. Homophily has no qualitative effect on paradigm shifts

Homophily is considered an important mechanism for cultural spreading [31,35]. It is the basis of the dynamics of well-established models, like Axelrod's [31]. In spite of that, it does not seem to have any influence on the occurrence of paradigm shifts, according to our simulations.

We have introduced the effect of homophily in the probability that a trait is adopted through the exponent $\alpha$ in function (2.6). This function gauges the Hamming distance (common traits) of the most influential people. For a given $\alpha$, this distance is $d_H(\mathbf{s}, \mathbf{s}') = n/(1 + \alpha)$. Thus, for $\alpha = 0$ (the value adopted so far) $\mathcal{W}(\mathbf{s}, \mathbf{s}')$ measures the probability that the trait in discussion is different in both individuals, regardless of the similarity between their cultural states. Accordingly, the most influential people are the most dissimilar ones because in an interaction, the fewer the traits the interacting individuals agree upon, the higher the probability that one of them is 'discussed' and a change ensues. If we set $\alpha = 1$, function $\mathcal{W}(\mathbf{s}, \mathbf{s}')$ is also proportional to the number of common traits of the two interacting individuals. This renders people with half the traits in common the most influential ones. Larger values of $\alpha$ strengthen this effect, so that when $\alpha = n - 1$ (that is $\alpha = 5$ in our case) the most influential individuals are those with just a single different trait.

Figure 7 shows similarity curves for a population evolving in the landscape of figure 2b, for three values of $\alpha$ (=0, 1 and 5). Despite the different behaviour of the similarity curves all three exhibit abrupt paradigm shifts—if only for slightly larger values of $\tau$ when $\alpha > 0$.

## 4. Discussion and conclusion

Human history exhibits long periods of cultural stasis punctuated by sudden changes that drastically transformed the prevailing paradigm. Many of these paradigm shifts were driven by major environmental or cognitive changes [9,39], or by crucial inventions that transformed the way humans obtained resources from the environment or made profit to improve well-being [11,12]. Nonetheless

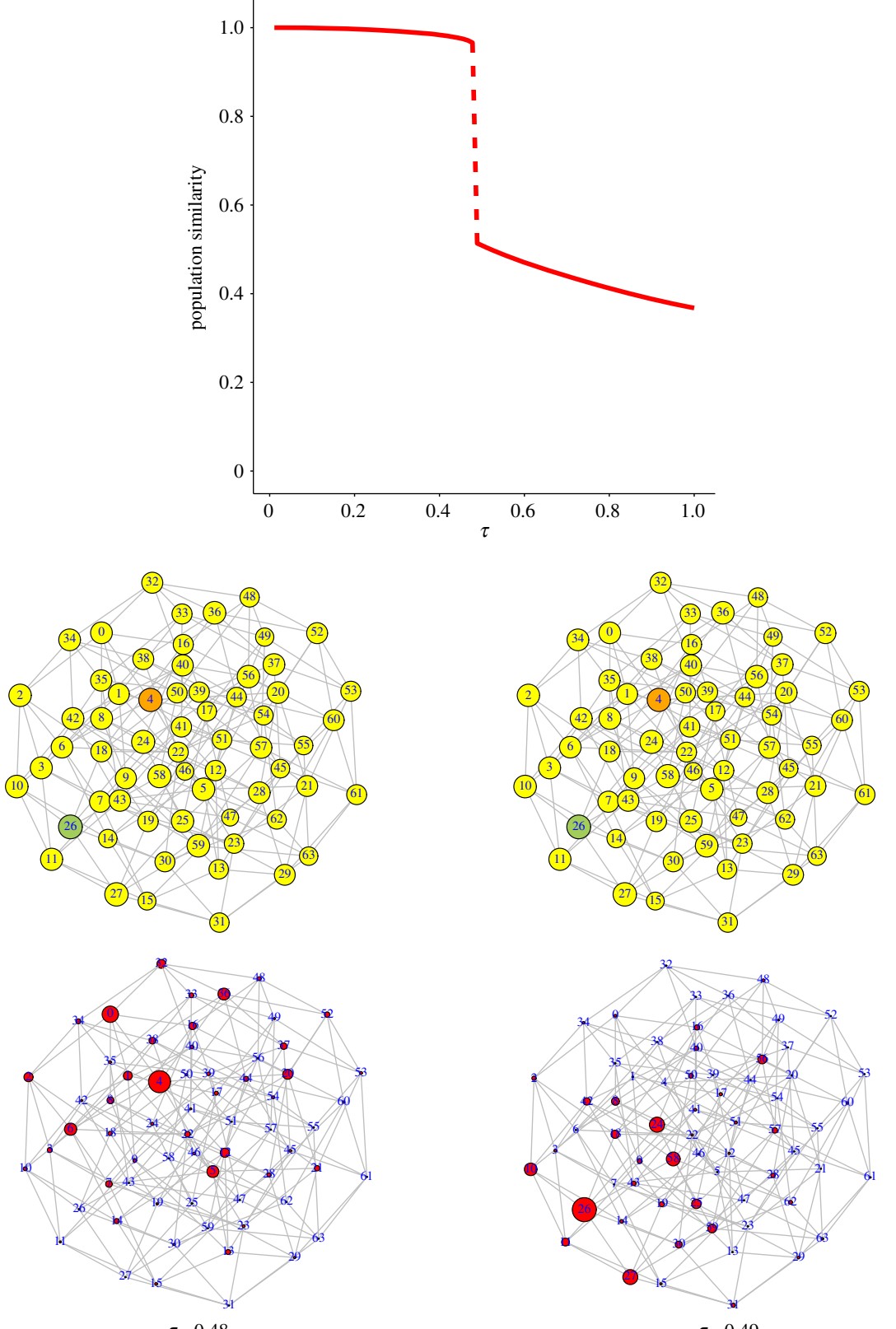

**Figure 6.** Similarity curve for the landscape of figure 2*b* with $\beta = 0.1$. Decreasing the value of $\beta$ increases the tolerance to inconsistencies, leading to quantitatively smaller jumps that occur at smaller values of $\tau$.

there are historical transitions that are more difficult to explain in these terms. For instance, it is not at all clear which critical breakthroughs underlie the transition from the Middle Ages to the Renaissance with which we introduced this article. In these cases, the paradigm shift seems better described as a change in

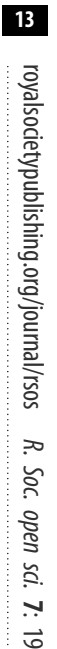

**Figure 7.** Effect of homophily in cultural paradigm shifts. The similarity curve for the landscape of figure 2b and different degrees of homophily is shown. (a) Homophily is absent in the transmission mechanism ($\alpha = 0$), (b) transmission is mildly influenced by homophily ($\alpha = 1$), and (c) transmission is strongly influenced by homophily ($\alpha = 5$). In (a), the most influential people are those with the least number of traits in common (because that maximizes the number of traits subject to an eventual change). In (b), people having half the traits in common are the most influential. In (c), changes are mainly driven by people with a single different trait. Despite these differences—and admitting that the specific dynamics changes with and without homophily—paradigm changes occur in all three cases.

beliefs, attitudes, customs, etc., perhaps as a consequence of new challenges. At a different scale, changes in the *zeitgeist* fall in the same category.

Our model is an attempt to explain paradigm shifts as an interplay between maintaining a coherent cultural state and coping with an environment that smoothly but unavoidably changes. Its main result is to reveal that changes are sudden and abrupt only if cultural epistasis is taken into account. Cultural and biological evolution share many common ideas and mechanisms, even though they also differ in many details. However, epistasis seems to be as relevant in cultural as it is in biological evolution [40]. The existence of correlations among traits create cultural states whose internal coherence is maximal with respect to changes in single traits. These cultural states represent equilibria of the dynamics that result from a complex interplay between fitness, landscape topology and population dynamics, such that they cannot be ascribed to any of the previous variables alone. A population 'trapped' in such coherent states will have it difficult to evolve unless the interaction between traits changes as a consequence of exogenous, imperceptible causes. But then a cascade of trait changes can lead to a new, more coherent cultural state. This is the microscopic description that our model provides of a paradigm shift. The same mechanism might apply to sudden transitions observed in other systems

where major causes cannot be adduced. For example, it could be argued that correlations in the microscopic features defining language evolution [22,41,42] may translate into languages evolving through punctuational bursts [43] or be responsible for occasionally fast linguistic evolution, as the coming of age of the English language at the beginning of the nineteenth century [44].

The model is deliberately simple because it is proposed as a proof of concept. To begin with, it assumes an infinite population, a common and reasonable assumption when studying the evolution of individual traits [16]. However, the number of cultural states diverges exponentially with the number of traits (even for six traits this number is 64), so that a population must be really huge for its fractions $x(\mathbf{s})$ to be meaningful. Otherwise, demographic noise becomes relevant and the model requires either a stochastic treatment or agent-based simulations. The model is a mean-field model that assumes well-mixed populations. An extension of the model could explicitly introduce space, limiting the interaction between individuals to close neighbours. In general, space-explicit representations lead to longer transients of higher diversity due to the appearance of local clustering. This is an interesting (but computationally highly costly) avenue to explore the emergence of metastable states, multi-stability and hysteresis. Another simplifying assumption is that cultural transmission is only horizontal. So far all individuals are contemporaries and never reproduce. Also their learning rate is constant in time and uniform in the population. We have added a small fraction of spontaneous changes in the trait of an individual to the model, mainly to avoid the disappearance of cultural states. We have checked that if the ratio between spontaneous change and transmission increases above ~1%, transmission becomes irrelevant and the population simply reflects the fitness landscape. We are not aware of any empirical data regarding those rates that can corroborate our choice but, intuitively, it looks, if anything, like an overestimation of the true rate. Finally, we have used one of the simplest models that contain epistasis to recreate the fitness landscape. Being more precise in the choice of fitness would amount to specifying what the cultural traits are (a complex endeavour in itself [45]) and figuring out a model that described how they interact with each other.

In spite of all these assumptions, the occurrence of drastic paradigm shifts appears to be very robust to the particulars of the model. To be precise, paradigm shifts appear regardless of the way epistasis is introduced (either through the NK landscape parameters or through trait incompatibility), of the consideration of homophily, and even of the degree of cognitive dissonance that incompatible traits may bring about to the individuals. To keep the simulations computationally tractable, we have restricted them to a low number of traits $n$, but have explored values of $k$ up to $n-1$. Changing the values of $k$ does not entail any qualitative change. In its turn, larger values of $n$ or the study of finite populations could probably increase the frequency and sharpness of the shifts, as it has been shown to occur in models of molecular evolution [20,21]. These results notwithstanding, all these modifications entail measurable quantitative changes.

An interesting prediction of the model is that the end state depends on the initial state, so that different populations, exposed to the same environment, may give rise to different cultures. This result stems from the nature of cultural transmission: the likelihood that two individuals with different cultural states meet depends on the fraction of population in each state. This entails frequency-dependent selection (compare e.g. with the model of [20] for molecular evolution) and, as it happens in analogous processes in biology (e.g. if recombination is considered) the equilibrium state is not unique [46]. In such systems, irreversibility is common. Indeed, once a paradigm shift happens, it cannot be reverted by simply restoring the external conditions back to their primitive values. It is very difficult to illustrate this effect with real-life situations or historical events, mainly because other mechanisms may be at play simultaneously. However, this could be one of the predictions that might allow an empirical validation of our model. Though figuring out an experiment that can directly test the assumptions of the model seems hard (guessing a fitness landscape from the interactions between different traits looks, at this point, hopeless), devising a situation in which some external influence is first changed and later restored, and measuring how this affects the emergent state in a population looks feasible. For example, it has been shown that minority groups can initiate social change dynamics and lead to the emergence of new social conventions [47]. The influence of minority groups could be easily reversed in that environment. If the backwards pathway to the previous convention differs from its forward realization, this might provide an indirect test of the model predictions, and give support to the sensible expectation of irreversibility of cultural paradigm shifts.

Data accessibility. Relevant code for this research work is stored in GitHub: https://github.com/Ignacio-Pascual/cultural_dynamics and has been archived within the Zenodo Repository: https://doi.org/10.5281/zenodo.3614021 [48].

Authors' contributions. J.A.C. designed the model, I.P. carried out the simulations, I.P., J.A., S.M. and J.A.C. discussed the results and designed the research, I.P., S.M. and J.A.C. wrote the paper.

Competing interests. We declare we have no competing interest.

Funding. This work was supported by the Spanish projects VARIANCE (FIS2015-64349-P, MINECO/FEDER, UE), BASIC (FIS2018-098186-B-I00, MICINN/FEDER, UE), MiMevo (FIS2017-89773-P, MINECO/FEDER, UE) and SEV-2013-0347 (MINECO).

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
