## [Reviewer comments · Royal Society Open Science]

Review History

RSOS-191813.R0 (Original submission)

Review form: Reviewer 1

Is the manuscript scientifically sound in its present form?

Yes

Are the interpretations and conclusions justified by the results?

Yes

Is the language acceptable?

Yes

Do you have any ethical concerns with this paper?

No

Have you any concerns about statistical analyses in this paper?

No

Recommendation?

Major revision is needed (please make suggestions in comments)

Comments to the Author(s)

In this work the authors present a model for paradigms shift along cultural evolution, based on Axelrod' s original proposal for cultural evolution.

The main conclusion in this work is that epistasis, properly defined in the context of the present model, is responsible for these shifts.

The work explores original ideas and present a novel formal model, that although based on previous works, contains interesting new ideas.

The presentation of the model, the mathematical approach and results are presented in a clear way but I have some observation listed below i would ask the authors to address

1) Fig 1 is not clear. In the text the authors say that figs. a and c show the fitness and b and d the cultural state, while the legend in the vertical axis reads "fitness" in all cases. The number of nodes seems not to be the same, not even when considering the pairs and b, c and d.

By the definition, the cultural states can be only 0 or 1, while the plot shows something different, perhaps due to an issue with the graphing soft.

2) It could be a bit uncomfortable reading about "fitness" without a proper definition of the concept until much later.

I understand that the organization of the manuscript might required the fitness to be introduced later, but perhaps the authors could find a solution not to delay the presentation of the concept so much.

3) There seems to be an inconsistency in the definition of dH. The first definition states that it is the usual Hamming distance, i.e. the number of differences in the values of the cultural traits between two nodes. But later the authors say that the maximum value of dH is reached for a given value of alpha. How is that the Hamming distance depends on alpha if it only measures the number of coincidences? At least this not clear considering the role assigned to alpha

4) The nodes are numbered and ordered accordingly. If we locate the nodes in the ring, the epistasis, when present, links each node with its neighbours in only one direction and always in order. Is this restriction something that could affect the results? It is hard to accept that there is no reciprocity in the epistasis unless the value of k is close to the value of N.

5) The fitness, that is governing the evolution of the cultural profile of the population, is not affected by homophily so why should homophily be important? Is not that a trivial results?

Review form: Reviewer 2

Is the manuscript scientifically sound in its present form?

Yes

Are the interpretations and conclusions justified by the results?

Yes

Is the language acceptable?

Yes

Do you have any ethical concerns with this paper?

No

Have you any concerns about statistical analyses in this paper?

No

Recommendation?

Accept with minor revision (please list in comments)

Comments to the Author(s)

In the manuscript entitled "Epistasis between cultural traits cause paradigm shifts in cultural evolution" by Ignacio Pascual et al. the authors develop a simple model for cultural dissemination showing that paradigm shifts can occur as emergent phenomena of the underlying dynamics.

In this model, the individuals are characterized by a vector of cultural traits that change mainly through cultural contagion biased by "cultural fitness" landscape. The authors show that this model produces paradigm shifts in response to weak changes in the landscape only in the presence of cultural epistasis.

The model is perfectly framed and its equations are rigorously derived. The presented results are clear and informative and the discussion highlights the virtues of the model, but also its limitations and possible future works.

The manuscript is well written and, from my point of view, it deserves to be published in RSOS, after the authors can clarify some minor comments and questions:

1 - In line 14 of page 5, the sentence reads "The last term in this expression" referring to equation (2.5). However, this equation has only one term.

2 - The description of the Kauffman's NK landscape is a little bit confusing for me. Could the authors improve this section? Here, in line 27 which describes equation (2.10) they refer to an index $|j|$ which seems to be absent in the mentioned equation.

3 - I do not understand figure 1. Four panels refer to fitness as a function of tau for all nodes? Which are the differences between panels (a) and (b) (and between (c) and (d)?). They correspond to initial and final times of evolution? The description either in the text or in the caption is not enough to understand this figure.

4 - In line 45 of page 8, when comparing figures 2 and 3, they refer to figure 2 (c). Should be Figure 2 (b) ?

Decision letter (RSOS-191813.R0)

14-Jan-2020

Dear Dr Cuesta

On behalf of the Editors, I am pleased to inform you that your Manuscript RSOS-191813 entitled "Epistasis between cultural traits causes paradigm shifts in cultural evolution" has been accepted for publication in Royal Society Open Science subject to minor revision in accordance with the referee suggestions. Please find the referees' comments at the end of this email.

The reviewers and handling editors have recommended publication, but also suggest some minor revisions to your manuscript. Therefore, I invite you to respond to the comments and revise your manuscript.

- Ethics statement

- Data accessibility

<http://datadryad.org/submit?journalID=RSOS&manu=RSOS-191813>

- Competing interests

- Authors' contributions

- Acknowledgements

- Funding statement

Please ensure you have prepared your revision in accordance with the guidance at <https://royalsociety.org/journals/authors/author-guidelines/> -- please note that we cannot

publish your manuscript without the end statements. We have included a screenshot example of the end statements for reference. If you feel that a given heading is not relevant to your paper, please nevertheless include the heading and explicitly state that it is not relevant to your work.

Because the schedule for publication is very tight, it is a condition of publication that you submit the revised version of your manuscript before 23-Jan-2020. Please note that the revision deadline will expire at 00.00am on this date. If you do not think you will be able to meet this date please let me know immediately.

If your manuscript is newly submitted and subsequently accepted for publication, you will be asked to pay the article processing charge, unless you request a waiver and this is approved by Royal Society Publishing. You can find out more about the charges at <https://royalsocietypublishing.org/rsos/charges>. Should you have any queries, please contact openscience@royalsociety.org.

on behalf of Professor Matjaz Perc (Associate Editor) and Miles Padgett (Subject Editor)
openscience@royalsociety.org

Reviewer comments to Author:
 Reviewer: 1

Comments to the Author(s)

In this work the authors present a model for paradigms shift along cultural evolution, based on Axelrod's original proposal for cultural evolution. The main conclusion in this work is that epistasis, properly defined in the context of the present model, is responsible for these shifts.

The work explores original ideas and present a novel formal model, that although based on previous works, contains interesting new ideas.

The presentation of the model, the mathematical approach and results are presented in a clear way but I have some observation listed below i would ask the authors to address

1) Fig 1 is not clear. In the text the authors say that figs. a and c show the fitness and b and d the cultural state, while the legend in the vertical axis reads "fitness" in all cases. The number of nodes seems not to be the same, not even when considering the pairs and b, c and d.

By the definition, the cultural states can be only 0 or 1, while the plot shows something different, perhaps due to an issue with the graphing soft.

2) It could be a bit uncomfortable reading about "fitness" without a proper definition of the concept until much later.

I understand that the organization of the manuscript might required the fitness to be introduced later, but perhaps the authors could find a solution not to delay the presentation of the concept so much.

3) There seems to be an inconsistency in the definition of dH. The first definition states that it is the usual Hamming distance, i.e. the number of differences in the values of the cultural traits between two nodes. But later the authors say that the maximum value of dH is reached for a given value of alpha. How is that the Hamming distance depends on alpha if it only measures the number of coincidences? At least this not clear considering the role assigned to alpha

4) The nodes are numbered and ordered accordingly. If we locate the nodes in the ring, the epistasis, when present, links each node with its neighbours in only one direction and always in order. Is this restriction something that could affect the results? It is hard to accept that there is no reciprocity in the epistasis unless the value of k is close to the value of N .

5) The fitness, that is governing the evolution of the cultural profile of the population, is not affected by homophily so why should homophily be important? Is not that a trivial results?

Reviewer: 2

Comments to the Author(s)

In the manuscript entitled "Epistasis between cultural traits cause paradigm shifts in cultural evolution" by Ignacio Pascual et al. the authors develop a simple model for cultural dissemination showing that paradigm shifts can occur as emergent phenomena of the underlying dynamics.

In this model, the individuals are characterized by a vector of cultural traits that change mainly through cultural contagion biased by "cultural fitness" landscape. The authors show that this model produces paradigm shifts in response to weak changes in the landscape only in the presence of cultural epistasis.

The model is perfectly framed and its equations are rigorously derived. The presented results are clear and informative and the discussion highlight the virtues of the model, but also its limitations and possible future works.

The manuscript is well written and, from my point of view, it deserve to be published in RSOS, after the authors can clarify some minor comments and questions:

1 - In line 14 of page 5, the sentence reads "The last term in this expression" referring to equation (2.5). However, this equation have only one term.

2 - The description of the Kauffman's NK landscape is a little bit confusing for me. Could the authors improve this section? Here, in line 27 which describe equation (2.10) they refer to an index $|j|$ which seems to be absent in the mentioned equation.

3 - I do not understand figure 1. Four panels refers to fitness as a function of tau for all nodes? Which are the difference between panels (a) and (b) (and between (c) and (d)?). They correspond to initial and final times of evolution? The description either in the tex or in the caption are not enough to understand this figure

4 - In line 45 of page 8, when comparing figures 2 and 3, the refer to figure 2 (c). Should be Figure 2 (b) ?

Author's Response to Decision Letter for (RSOS-191813.R0)

See Appendix A.

Decision letter (RSOS-191813.R1)

30-Jan-2020

Dear Dr Cuesta,

It is a pleasure to accept your manuscript entitled "Epistasis between cultural traits causes paradigm shifts in cultural evolution" in its current form for publication in Royal Society Open Science. The comments of the reviewer(s) who reviewed your manuscript are included at the foot of this letter.

on behalf of Professor Matjaz Perc (Associate Editor) and Miles Padgett (Subject Editor)
openscience@royalsociety.org

Associate Editor Comments to Author (Professor Matjaz Perc):

Associate Editor

Comments to the Author:

Thank you for the comprehensive revision of your manuscript, which we are happy to accept for publication in Royal Society Open Science.

Reviewer comments to Author:

Appendix A

Dear Editor of Royal Society Open Science,

Thank you very much for your accepting our manuscript for publication in Royal Society Open Science. We have gone through all minor comments that the referees made and either fixed them or provided a response to them all. In what follows you will find the detailed responses.

Best wishes,

J. A. Cuesta, on behalf of all authors

Reviewer comments to Author:

Reviewer: 1

Comments to the Author(s)

In this work the authors present a model for paradigms shift along cultural evolution, based on Axelrod's original proposal for cultural evolution.

The main conclusion in this work is that epistasis, properly defined in the context of the present model, is responsible for these shifts.

The work explores original ideas and present a novel formal model, that although based on previous works, contains interesting new ideas.

The presentation of the model, the mathematical approach and results are presented in a clear way but I have some observation listed below i would ask the authors to address

We thank the referee for the very positive overall judgment and for the careful revision.

1) Fig 1 is not clear. In the text the authors say that figs. a and c show the fitness and b and d the cultural state, while the legend in the vertical axis reads "fitness" in all cases. The number of nodes seems not to be the same, not even when considering the pairs and b, c and d.

There y axis in b and d was incorrectly labeled, thank you for spotting the error: it has to be population density. All simulations have 64 nodes (2^6). The figure has been substituted by its raw, unedited version for clarity. Here, any impression of a different number of nodes is due to the 2D projection.

By the definition, the cultural states can be only 0 or 1, while the plot shows something different, perhaps due to an issue with the graphing soft.

The reviewer is right that each trait takes values 0 or 1. But the cultural state is defined by a vector of such traits which is assigned a real fitness value between 0 and 1. We have revised the description of the model and of the variables involved and have found no mistake that could lead to that confusion. Therefore, we have not changed the ms in regard to that comment.

2) It could be a bit uncomfortable reading about "fitness" without a proper definition of the concept until much later.

I understand that the organization of the manuscript might required the fitness to be introduced later, but perhaps the authors could find a solution not to delay the presentation of the concept so much.

We have moved previous subsections introducing the definition of fitness landscapes and Kauffman's NK model; they precede now the definition of the system dynamics. The last paragraph in the latter section has been also moved for consistency and an introductory sentence (highlighted in red) has been added.

3) There seems to be an inconsistency in the definition of dH . The first definition states that it is the usual Hamming distance, i.e. the number of differences in the values of the cultural traits between two nodes. But later the authors say that the maximum value of dH is reached for a given value of α . How is that the Hamming distance depends on α if it only measures the number of coincidences? At least this not clear considering the role assigned to α

We are unable to spot the inconsistency stated by the referee. Indeed, dH is the usual Hamming distance. The maximum we talk about corresponds to a complex function of different parameters and variables, among them dH and the degree of homophily. And this function has a maximum when certain combination of dH and α occurs. This does not imply that dH depends on α since, as the referee states, it just measures the number of coincidences. This nonetheless, we have rephrased the sentence to clarify that it is not the increase in α that directly modifies the distance between individuals.

4) The nodes are numbered and ordered accordingly. If we locate the nodes in the ring, the epistasis, when present, links each node with its neighbours in only one direction and always in order. Is this restriction something that could affect the results? It is hard to accept that there is no reciprocity in the epistasis unless the value of k is close to the value of N .

Kauffman's NK model is admittedly a cartoon model for real epistasis. However, we use it because its qualitative properties correctly capture the gross features of epistasis. In a cultural context, data are insufficient to describe even the most generic properties of the landscape. The reciprocity the reviewer mentions holds in general in molecular sequences, but perhaps this is not so in cultural traits. For example, being vegetarian implies that a person does not eat meat, but not eating meat can depend on other personal choices. Despite the lack of reciprocity implicit in the definition of the landscape, the fact that the NK model is able to quantitatively mimic the landscape of RNA secondary structure, for example, strongly supports that this restriction does not affect the results. (Note that nodes are linked to the same number of neighbors in both directions, as explained right after Eq. (2.2) – (2.10) in the previous version. Whether symmetrically linked, as done, or always in one direction and in order, is however irrelevant for the results.)

5) The fitness, that is governing the evolution of the cultural profile of the population, is not affected by homophily so why should homophily be important? Is not that a trivial results?

Homophily affects the distribution of the population in the fitness landscape, and it is in this sense that it might be important: it changes the cultural diversity and often clusters the population in separated groups. Still, it does not affect the occurrence of paradigm shifts. In our view, this result is not trivial. On the contrary, since homophily is an important factor in cultural spread, it is important to show that its presence does not condition the occurrence of such shifts.

Reviewer: 2

Comments to the Author(s)

In the manuscript entitled “Epistasis between cultural traits cause paradigm shifts in cultural evolution” by Ignacio Pascual et al. the authors develop a simple model for cultural dissemination showing that paradigm shifts can occur as emergent phenomena of the underlying dynamics.

In this model, the individuals are characterized by a vector of cultural traits that change mainly through cultural contagion biased by “cultural fitness” landscape.

The authors show that this model produces paradigm shifts in response to weak changes in the landscape only in the presence of cultural epistasis.

The model is perfectly framed and its equations are rigorously derived. The presented results are clear and informative and the discussion highlight the virtues of the model, but also its limitations and possible future works.

We thank the referee for the very positive overall judgment and for the careful revision.

The manuscript is well written and, from my point of view, it deserve to be published in RSOS, after the authors can clarify some minor comments and questions:

1 – In line 14 of page 5, the sentence reads “The last term in this expression” referring to equation (2.5). However, this equation have only one term.

We have changed “term” by “factor”, since we meant the last “multiplicative term”.

2 – The description of the Kauffman’s NK landscape is a little bit confusing for me. Could the authors improve this section? Here, in line 27 which describe equation (2.10) they refer to an index $|j|$ which seems to be absent in the mentioned equation.

Though there is some care required in the generations of fitness landscapes through Kauffman’s NK model, there are abundant examples that the reader can easily find. Therefore, we would prefer to keep this section (which is however self-contained) to a minimum, as it is.

The line below (former) equation (2.10) describes the meaning of the notation $[j]$ for any value of j . We refer to an “abstract” index j because in the equation we use $[i+1]$, $[i+2]$, etc., and the definition applies to them all. For instance, if $i+3$ happens to be no larger than n , then $[i+3]=i+3$, whereas if $i+3>n$, then $[i+3]=i+3-n$. As a matter of fact, $[j]$ is introduced as a shorthand for $1 + (j \bmod n)$ – which would definitely be more cumbersome to use in the expressions.

3 – I do not understand figure 1. Four panels refers to fitness as a function of tau for all nodes? Which are the difference between panels (a) and (b) (and between (c) and (d)?). They correspond to initial and final times of evolution? The description either in the text or in the caption are not enough to understand this figure

We apologize for the mistake in the labels of axis y in panels (b) and (d), as spotted by the first referee and corrected in the new figure: these panels represent populations. Hopefully the representation is now clear. Also, the caption has been extended to better explain what is represented in each panel.

4 – In line 45 of page 8, when comparing figures 2 and 3, the refer to figure 2 (c). Should be Figure 2 (b)?

The reference to figure 2 (c) is correct. Note that figure 3 only represents the transitions in Fig. 2 (a) and (b), while the sudden shift (which is present in Fig. 2 (b), but not in Fig. 2 (a)) is qualitatively analogous to the transition in Fig. 2 (c) (not shown in Fig. 3).